# Abnormal outer hair cell efferent innervation in *Hoxb1*-dependent sensorineural hearing loss

Maria Di Bonito[1]*, Jérôme Bourien[2], Monica Tizzano[3], Anne-Gabrielle Harrus[2], Jean-Luc Puel[2], Bice Avallone[3‡], Regis Nouvian[2‡], Michèle Studer[1‡]*

1 Université Côte d'Azur (UCA), CNRS, Inserm, Institute of Biology Valrose (iBV), Nice, France, 2 University of Montpellier, Inserm, CNRS, Institute for Neurosciences of Montpellier (INM), Montpellier, France, 3 University of Naples Federico II, Department of Biology, Naples, Italy

‡ These authors are co-last senior authors on this work.
* marydibon71@gmail.com (MDB); Michele.STUDER@univ-cotedazur.fr (MS)

**Data Availability Statement:** All relevant data are within the manuscript and its Supporting Information files.

## Abstract

Autosomal recessive mutation of *HOXB1* and *Hoxb1* causes sensorineural hearing loss in patients and mice, respectively, characterized by the presence of higher auditory thresholds; however, the origin of the defects along the auditory pathway is still unknown. In this study, we assessed whether the abnormal auditory threshold and malformation of the sensory auditory cells, the outer hair cells, described in *Hoxb1null* mutants depend on the absence of efferent motor innervation, or alternatively, is due to altered sensory auditory components. By using a whole series of conditional mutant mice, which inactivate *Hoxb1* in either rhombomere 4-derived sensory cochlear neurons or efferent motor neurons, we found that the hearing phenotype is mainly reproduced when efferent motor neurons are specifically affected. Our data strongly suggest that the interactions between olivocochlear motor neurons and outer hair cells during a critical postnatal period are crucial for both hair cell survival and the establishment of the cochlear amplification of sound.

## Author summary

Hearing depends on sound stimuli transduction in the inner ear as well as sensory information processing within the central nervous system through functional ascending and descending auditory pathways. In humans, autosomal recessive mutation of *HOXB1* causes sensorineural hearing loss characterized by the elevation of auditory thresholds. Although this defect could be reproduced in a genetically modified mouse model in which the mouse *Hoxb1* gene was constitutively inactivated, the cellular origin underlying *HOXB1*-related deafness was still unknown. Here, we used a series of conditional mouse models to independently inactivate *Hoxb1* either in sensory neurons or in motor neurons. By combining electrophysiological, anatomical, and molecular analyses, we show that *Hoxb1*-related deafness is partially reproduced only when motor neurons are severely reduced or lost during development and prior to hearing onset. Our data support the hypothesis that the physical interaction between motor neurons and hair cells in the

**Funding:** This work was funded by the "Agence Nationale Recherche" (ANR-15-CE15-0016-01) to M.S and J.L.P, and "Fondation Pour l'Audition" (FPA RD-2017-5) to M.S. Both funders fully supported this study and M.D.B. salary. The funders had no role in study design, data collection and analysis, decision to publish, or preparation of the manuscript.

**Competing interests:** The authors have declared that no competing interests exist.

cochlea during a critical developmental period is essential for the proper maturation and functioning of the sound cochlear amplification system.

## Introduction

Hearing relies on the mechano-transduction of acoustic cues by sensory hair cells in the cochlea of the inner ear and the neural message conveyance to the higher auditory centers through ascending pathways [1]. In the cochlea, outer hair cells (OHCs) amplify the sound stimulation while inner hair cells (IHCs) transduce sound stimuli into glutamate release onto afferent nerve fibers. The neural message is then conveyed *via* auditory afferent fibers to the cochlear nucleus (CN), ultimately reaching the auditory cortex [2]. In response, incoming afferent sensory auditory signals are modulated by efferent motor neurons, such as facial motor neurons (FMNs) that migrate caudally, and the lateral (LOC) and medial (MOC) olivo-cochlear neurons, originating from the embryonic inner ear efferent (IEE) precursors located in ventral rhombomere 4 of the developing hindbrain [3–6]. LOC motor neurons synapse with cochlear afferents just below the IHCs and control cochlear nerve excitability, participate in spatial sound localization and protect the cochlea from acoustic injury [7,8]. On the other hand, MOC neurons are innervated by reflex neurons of the posteroventral cochlear nucleus (PVCN) but also receive inhibitory GABAergic and glycinergic synaptic inputs, and inhibit cochlear OHC motility leading to a decrease in the sound cochlear amplification *via* the MOC reflex [9–15]. Another reflex, called the middle-ear muscle (MEM) reflex involves FMNs, which activate the stapedius muscles and reduce the amplitude of sound transmission through the middle ear [16–19]. Therefore, efferent motor neurons play an important role in modulating sound amplification, protecting the cochlea from noise-induced hearing damage, and reducing the masking effects of background noise [7,20–23].

The assembly of central and peripheral auditory subcircuits depends on the spatially and temporally ordered sequence of cell specification, migration, and connectivity during embryogenesis. Alterations to any one of these events can lead to abnormal circuit formation and consequent hearing impairments. Centrally, the majority of the auditory nuclei and their derivatives originate in the hindbrain or rhombencephalon, which becomes subdivided along the anteroposterior (AP) axis into repeated basic cellular units, called rhombomeres (r), and along the dorsoventral (DV) axis into subtype-specific domains that will generate distinct cell populations contributing to the adult brainstem organization [24]. Various cell lineage studies in mice have shown that auditory neurons are generated into distinct DV columns spanning r2-5 (reviewed in [25]).

Homeotic *Hox* genes, together with column-specific transcription factors, act in this process by imparting AP and DV positional identity, inducing cell-type differentiation and maturation, and orchestrating circuit formation [25–27]. Among *Hox* genes, *Hoxb1* plays an important role in the formation of different structures involved in the mouse auditory pathway [28]. Moreover, recessive mutations in the human *HOXB1* gene cause hereditary congenital facial paresis and sensorineural hearing impairments [29–32], suggesting an evolutionarily conserved key role for *Hoxb1/HOXB1* in setting up functional auditory circuits. Initially, mouse *Hoxb1* is expressed in the hindbrain with an anterior boundary between r4 and r3 (early expression), before becoming progressively restricted and maintained in r4 (late expression) [33–35]. Previous studies in mice have shown that *Hoxb1* is essential in imposing r4 identity along the AP axis of the developing hindbrain [28,36–39] and as a consequence, is required in the specification and assembly of r4-derived auditory sensory and motor neurons

[28]. Lineage tracing analysis using an r4-specific reporter gene has shown that r4 will contribute to various nuclei of the ascending sound pathway, such as the cochlear nucleus, the ventral nucleus of the lateral lemniscus, and the superior olivary complex [28,40]. Besides sensory neurons, in ventral r4 MOC, LOC, and FMN efferent neurons will form the sensorimotor reflex circuits needed for sound amplification and cochlea protection [5]. By genetically inactivating *Hoxb1*, constitutive *null* mutant mice have abnormally specified r4-derived sensory nuclei, lack IEE motor neurons, and exhibit malformed OHCs in the inner ear. These defects lead to defective sound amplification and increased auditory threshold [28] reproducing the hearing loss of patients with mutations in the *HOXB1* gene, who show a bilateral mild to moderate high-frequency hearing loss and no otoacoustic emissions, indicative of abnormal OHC functionality [29].

In this study, we aimed to understand the origin of the auditory defects due to the absence of *Hoxb1/HOXB1* by assessing the roles of sensory and/or motor components in the abnormal auditory threshold and OHC degeneration phenotypes described in *Hoxb1null* mutants. To this goal, we used a conditional genetic approach to independently inactivate *Hoxb1* either in the cochlear sensory neurons or in the efferent motor neurons and compared these mice to *Hoxb1null* mutant mice in which both motor and sensory neurons were impaired. By combining electrophysiological and anatomical analyses, we show that altered auditory thresholds observed in *Hoxb1null* mice are partially reproduced only when efferent motor neurons are severely reduced or lost during development and prior to hearing functionality. Our data support the hypothesis that the physical interaction between MOC neurons and OHCs during a critical developmental period is essential for the proper maturation and functioning of OHCs and cochlear amplification of sound.

## Results

### Loss of cochlear amplification in *Hoxb1null* mutant mice

To obtain a more detailed functional analysis of the hearing loss in the *Hoxb1null* adult mice, we probed the auditory brainstem response (ABR), as a proxy of the synchronous activation of nuclei along the ascending auditory pathway (**Fig 1A**), using tone bursts from 4 to 32 kHz. An 8-kHz tone burst elicited ABR is composed of multiples waves corresponding to the electrical responses of the auditory nerve (wave 1), the cochlear nucleus (wave 2), the superior olivary complex (wave 3), the lateral lemniscus (wave 4) and inferior colliculus (wave 5) (**Fig 1A and 1B**). Auditory thresholds in heterozygous mice (*Hoxb1HET*) were comparable to *Ctrl* mice (mean threshold: 27 ± 0.4 dB SPL vs 29.2 ± 1.2 dB SPL in *Ctrl* and *Hoxb1HET* mice respectively, $P = 0.33$, two-tailed Mann-Whitney Wilcoxon's test; **Fig 1B–1D and S1 Data**). In contrast, we observed elevated auditory thresholds in the *Hoxb1null* mice at all probed frequencies, i.e., across the tonotopic axis (mean threshold: 27 ± 0.4 dB SPL vs 66.2 ± 2.3 dB SPL in *Ctrl* and *Hoxb1null* mice respectively, $P = 0.014$, two-tailed Mann-Whitney Wilcoxon's test; **Fig 1B–1D and S1 Data**). These results suggest a non-operating amplification process within the cochlea, i.e., the loss of active mechanisms provided by OHC activity. Thus, to assess any defective cochlear amplification, we measured distortion product otoacoustic emissions (DPOAEs), which directly reflect the mechanical activity of OHCs. While we found a slight but significant reduction in DPOAEs of the *Hoxb1HET* mice compared to *Ctrl* mice (Mean $2f_1$-$f_2$ for f2 between 5 to 20 kHz: 23.5 ± 1.3 dB and 19.7 ± 0.6 dB in *Ctrl* and *Hoxb1HET* mice respectively, $P = 0.02$, two-tailed Mann-Whitney Wilcoxon's test; **Fig 1E and 1F and S1 Data**), DPOAEs from *Hoxb1null* mice were strongly reduced in comparison to *Ctrl* and *Hoxb1HET* mice (Mean $2f_1$-$f_2$ for f2 between 5 to 20 kHz: 4.3 ± 1.6 dB in *Hoxb1null* mice; $P = 6.10^{-4}$ in *Hoxb1null* vs *Ctrl* and $P = 1.10^{-4}$ in *Hoxb1null* vs *Hoxb1HET* mice, respectively, two-tailed

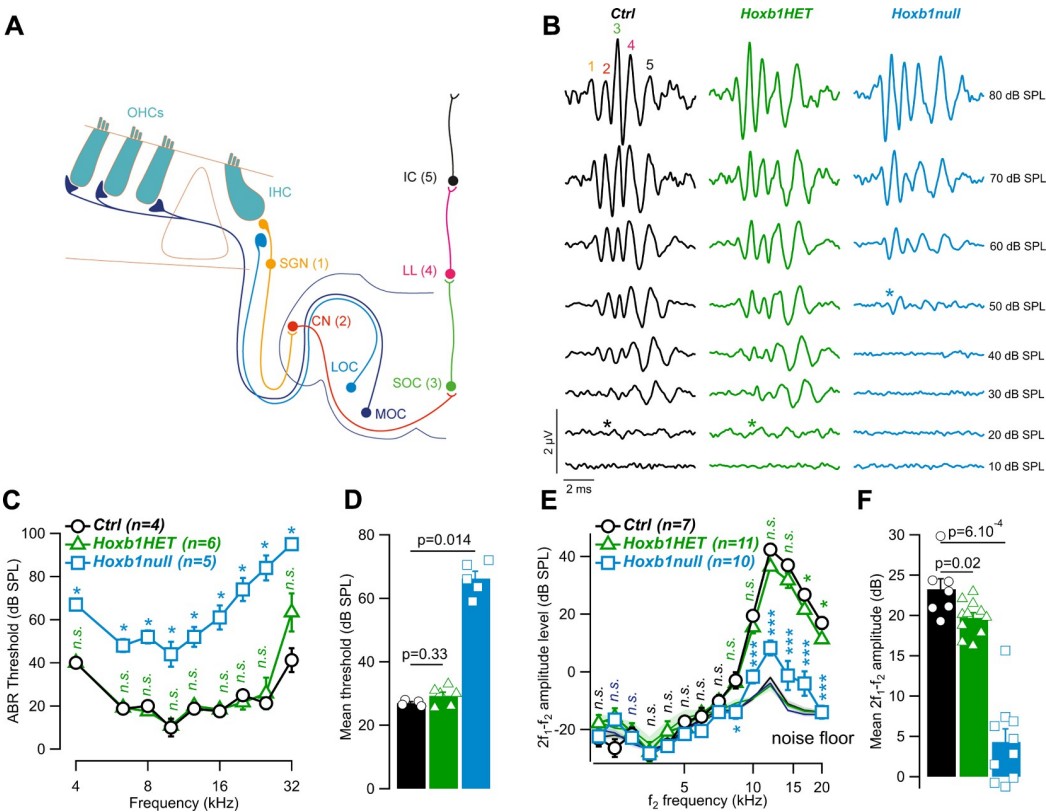

**Fig 1. Hearing loss in the *Hoxb1* constitutive loss-of-function mouse line.** (**A**) Schematic of the auditory pathway probed by the ABRs in (**B**). Each wave of the ABR reflects the synchronous activation of the nuclei along the ascending auditory pathway. (**B**) Representative example of ABR recordings evoked by 8-kHz tone burst in 1- to 3-month-old *Ctrl*, *Hoxb1HET* and *Hoxb1null* mice. (**C**) Mean ABR ± SEM audiograms from 1- to 3-month-old *Ctrl*, *Hoxb1HET* and *Hoxb1null* mice recorded from 10 to 100 dB SPL. (**D**) Mean threshold ± SEM from (**C**). (**E**) DPOAEs from 1- to 3-month-old *Ctrl*, *Hoxb1HET* and *Hoxb1null* mice. The $2f_1$-$f_2$ amplitude level is shown as a function of $f_2$ frequency. The black lines indicate the background noise level. (**F**) Mean $2f_1$-$f_2$ ± SEM amplitude level from (**E**) measured between 5 to 20 kHz. In (**C**) and (**E**), n indicates the number of cochleae recorded. Stars indicate the level of significance: *<0.05, ***<0.001 and n.s.>0.05, two-tailed Mann-Whitney Wilcoxon test. In (**D**) and (**F**), symbols represent individual cochleae. P indicates the level of significance, two-tailed Mann-Whitney Wilcoxon test. See also **S1** Data. Abbreviations: OHCs, outer hair cells; IHC, inner hair cell; MOC, medial olivocochlear neuron; LOC, lateral olivocochlear neuron; SGN, spiral ganglion neuron; CN, cochlear nucleus; SOC, superior olivary complex; LL, lateral lemniscus; IC, inferior colliculus.

Mann-Whitney Wilcoxon's test; **Fig 1E and 1F** and **S1 Data**). Together, our data show that loss of Hoxb1 function leads to defective OHC activity and thus to hearing impairment consistent with our previous work [28].

## Normal auditory thresholds and OHC morphology in r4-sensory conditional *Hoxb1* mutants

Given that r4-derivatives highly contribute to central sensory and motor components of the auditory system and that inactivation of *Hoxb1*, a key gene in r4 identity and specification, affects the development of all derivatives originating in r4 [28, 36, 37, 41], we decided to generate a series of *Hoxb1* conditional mice which would inactivate *Hoxb1* either in the sensory or motor cells of the auditory pathway (**Fig 2**). To this purpose, *Hoxb1flox/flox* mice [28] were first crossed to *Atoh1-Cre* [42,43] or *Ptf1a-Cre* [44,45] *knock-in recombinase* lines to specifically inactivate *Hoxb1* in glutamatergic or GABAergic/glycinergic cochlear neurons, respectively, since *Atoh1-* and *Ptf1a-*expressing neuroepithelial regions contribute to excitatory and

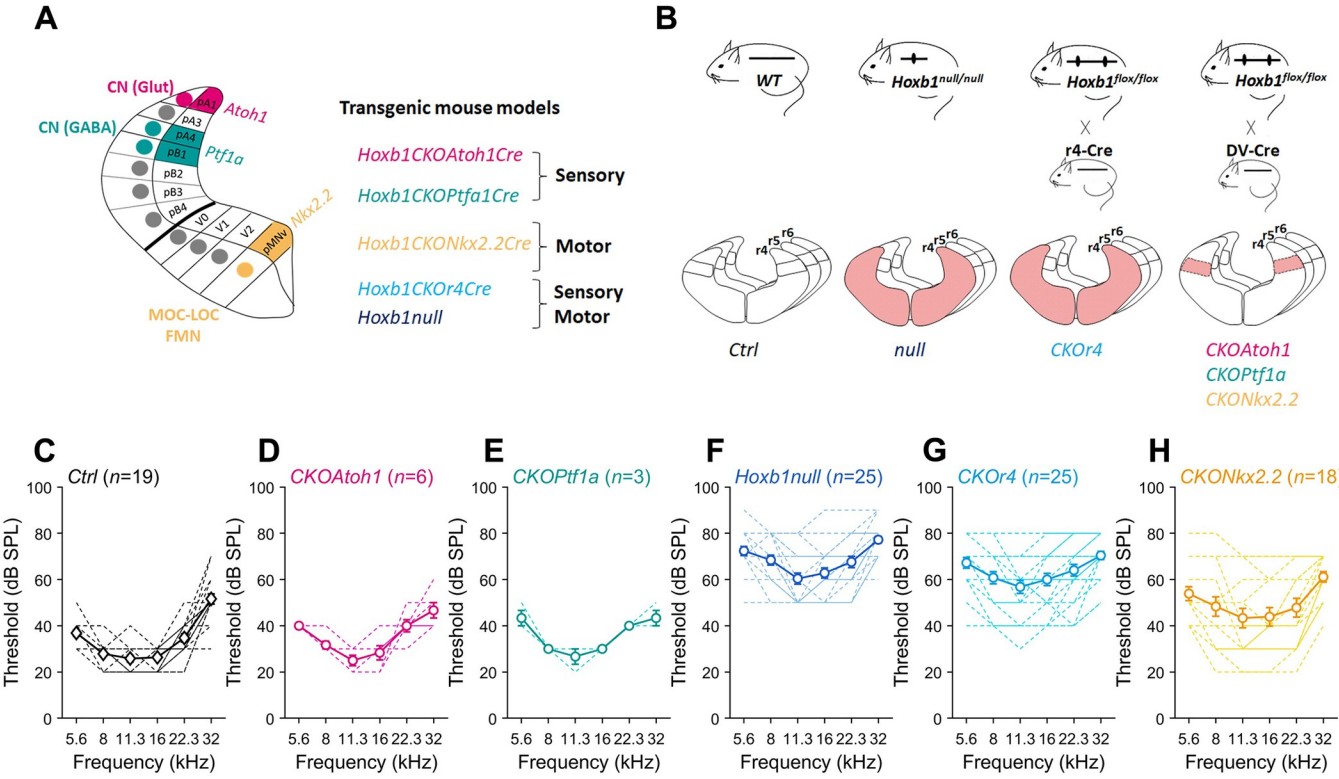

**Fig 2. Conditional *Hoxb1* inactivation in r4-derived motor and sensory neurons and recorded audiograms. (A)** Schematics of half of rhombomere 4 (r4) in the developing hindbrain showing DV domain of the activity of the different *Cre-recombinase* lines used in this study. While the *Nkx2.2-Cre* line is expressed in ventral r4 from which MOC, LOC, and FMN efferent motor neurons originated, the *Atoh1-* and *Ptf1-Cre* lines are expressed in distinct dorsal domains from which the sensory glutamatergic or GABAergic/glycinergic populations, respectively, of the cochlear nucleus (CN) are generated. **(B)** The genetic strategy used to inactivate *Hoxb1* in different domains. In the *null* group, *Hoxb1* is inactivated from the onset of expression; in the conditional *CKOr4* mice, *Hoxb1* expression is exclusively abolished in r4 from E8.5 onwards. DV-restricted inactivation is obtained by using the *CKOAtoh1* and *CKOPtf1a* mice for CN sensory components or the *CKONkx2.2* mice for the motor efferent structures. **(C-H)** Audiograms from distinct mouse groups are indicated above each graph. Empty circles: mean ABR ± SEM, dashed lines: individual cochleae. n indicates the number of cochleae recorded. See also **S1** Data.

inhibitory subtypes of the dorsal and ventral cochlear nuclei [4,46] (**Fig 2A and 2B**). These lines will be named respectively *Hoxb1CKOAtoh1Cre* (or *CKOAtoh1*) and *Hoxb1CKOPtf1aCre* (or *CKOPtf1a*) mutants throughout this study. In this way, the role of the cochlear sensory populations in sound transmission and hair cell survival could be directly investigated. Since *Hoxb1* is not expressed in hair cells in the cochlea [28,47] the *Atoh1-Cre* could not affect these populations. The *CKOAtoh1* and *CKOPtf1a* conditional mutant lines were compared with constitutive *Hoxb1null* (or *null*) or conditional *Hoxb1CKOr4Cre* (or *CKOr4*) in which *Hoxb1* is inactivated in the whole r4 domain at embryonic stages and thus abolished in both r4-derived motor and sensory neurons [28] (**Fig 2A and 2B**). While *null* mutants exhibit a complete loss of *Hoxb1* expression from its onset, only the late expression of *Hoxb1* in r4 is eliminated in *CKOr4* mice thanks to the use of the *b1r4-Cre* [28]. First, we probed the functional state of the auditory pathway in the *CKOAtoh1* and *CKOPtf1a* mice using ABR evoked by tone-burst. As shown in **Fig 2C–2E**, the audiograms were comparable between *Ctrl*, *CKOAtoh1* and *CKOPtf1a* mice (mean threshold: 33.9 ± 0.8 dB SPL, 35.3 ± 0.8 dB SPL and 35.6 ± 0.6 dB SPL in *Ctrl*, *CKOAtoh1* and *CKOPtf1a* mice respectively; *Ctrl* vs *CKOAtoh1*, P = 0.33; *Ctrl* vs *CKOPtf1a*, P = 0.33; *CKOAtoh1* vs *CKOPtf1a*, P = 0.88, two-tailed Mann-Whitney Wilcoxon's test; **S1 Data**). This suggests that the selective inactivation of *Hoxb1* in the sensory cochlear neurons does not alter auditory function.

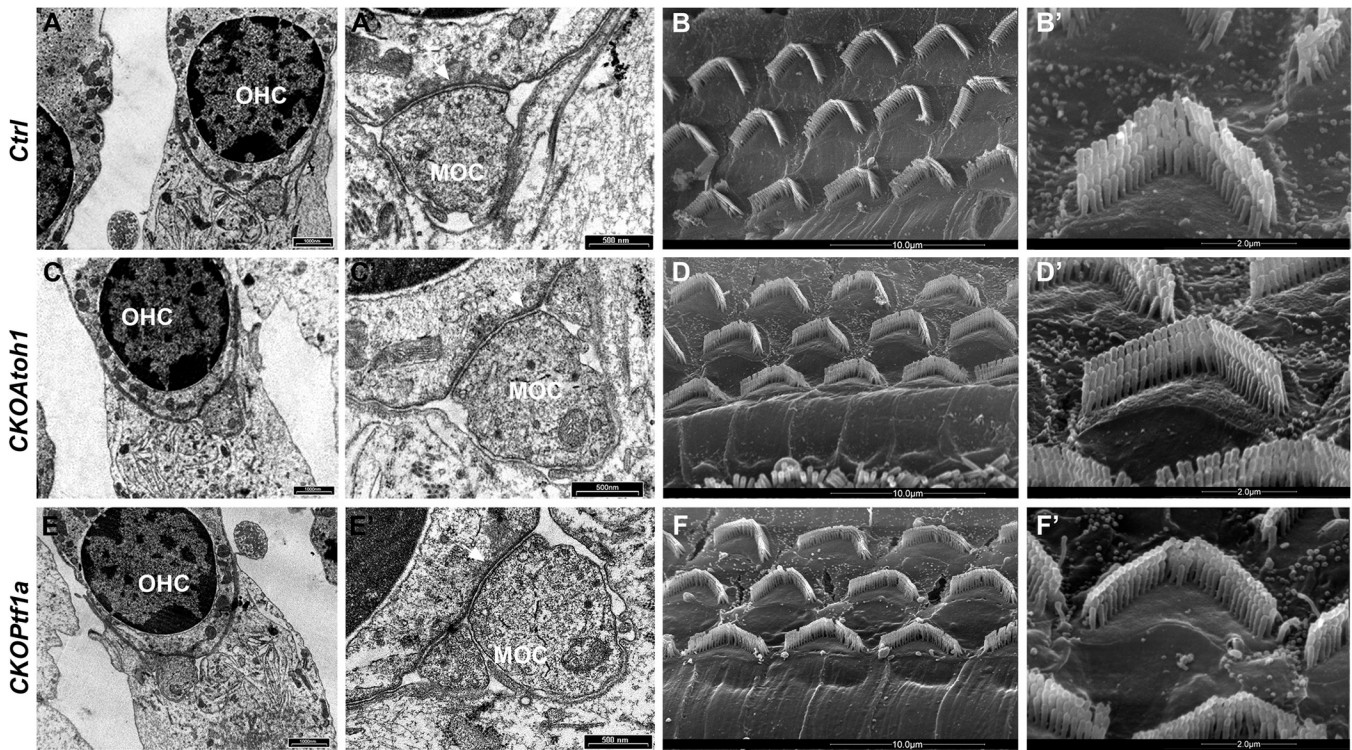

**Fig 3. Normal medial olivocochlear neuron (MOC) innervation and outer hair cell (OHC) morphology in r4-sensory *Hoxb1* mutant mice.** Transmission electron microscopy of MOC innervating OHC (**A, C, E**, high magnification in **A', C', E'**) and scanning electron microscopy of OHCs (**B, D, F**, high magnification in **B', D', F'**) in cochleae of *Ctrl* (A-B') and *Hoxb1* sensory conditional adult mice, *CKOAtoh1* (C-D') and *CKOPtf1a* (E-F'), in which *Hoxb1* is affected in glutamatergic and GABAergic/glycinergic cochlear sensory neurons, respectively. Scale bars 1000 nm (A, C, E), 500 nm (A', C', E'), 10 μm (B, D, F), 2 μm (B', D', F'). White arrows in A', C', E' point to synaptic cisterns at the interface between OHC and MOC.

To ensure that the normal threshold was accompanied by proper OHC morphology and efferent innervation, we harvested the cochleae of individual mice previously recorded in electrophysiological sessions and processed them for transmission electron microscopy (TEM) to detect the innervation of OHCs by MOC efferent neurons, and for scanning electron microscopy (SEM) to analyze the structure and organization of the OHCs in the cochlea (**Fig 3**). No significative differences in terms of the presence of MOC/OHC interactions were observed in *CKOAtoh1* and *CKOPtf1a* cochleae relative to control ones (**Fig 3A, 3A', 3C, 3C', 3E and 3E'**). In addition, the overall OHC row organization and V-shaped morphology were preserved in mutant cochleae (**Fig 3B, 3B', 3D, 3D', 3F and 3F'**). In all genotypes, MOC terminals show their typical structure with a well-recognized electron-dense cytoplasm characterized by the presence of synaptic cisterns (white arrows in **Fig 3A', 3C' and 3E'**), supporting a functional interaction between MOCs and OHCs, and in line with the normal auditory thresholds described in **Fig 2C–2E**. Taken together these data show that loss of Hoxb1 function in the excitatory and inhibitory cell populations of the cochlear nucleus does not affect proper OHC maturation and cochlear amplification of sound.

## Altered auditory thresholds in r4-motor conditional *Hoxb1* mutants

To assess whether the motor components originating from r4 were responsible for the auditory impairments observed in *Hoxb1null* mutants, we used the *Nkx2.2-Cre* line [48] to inactivate *Hoxb1* solely in ventral motor neurons (**Fig 2A and 2B**). As previously shown, the

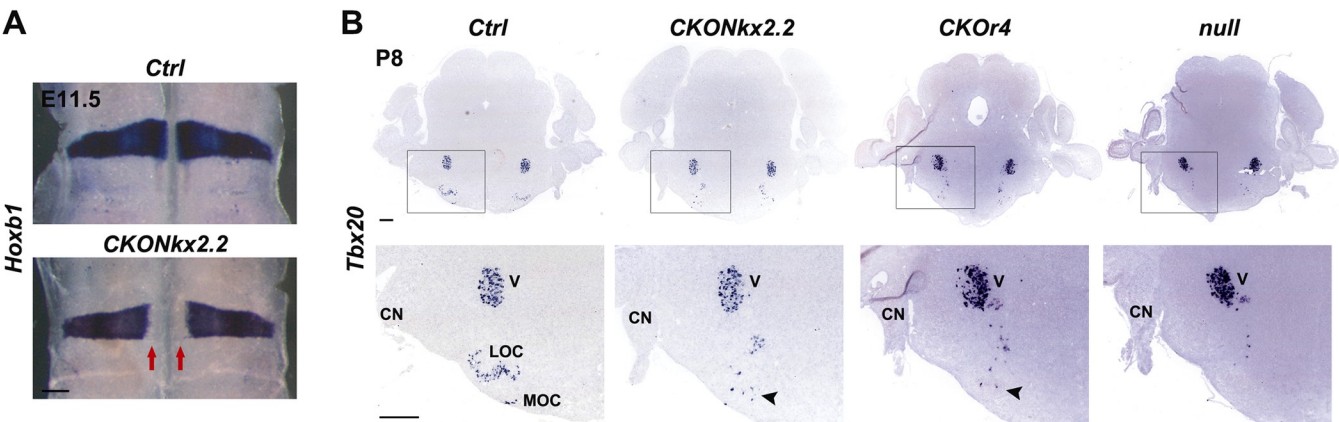

**Fig 4. Partial loss of olivocochlear neurons in r4-motor conditional *Hoxb1* mutant mice. A**) Dorsal views of E11.5 flat-mounted hindbrains stained with an antisense riboprobe for *Hoxb1*. Red arrows indicate *Hoxb1* loss solely in the r4 ventral motor domain of the *CKONkx2.2* mouse line. **B**) *In situ* hybridization of coronal sections of P8 brains positive for *Tbx20* labeling medial (MOC) and lateral (LOC) olivocochlear neurons and the trigeminal nucleus (V). Bottom panels show high-magnification images of boxed areas. Arrowheads point to the few spread cells in the ventral olivocochlear region of conditional mutant mice only. Ectopic dispersed *Tbx20*-expressing cells are present in more dorsal locations of all mutant mice. CN, cochlear nucleus. Scale bars 200 μm (A), 400 μm (B).

domain of ventral r4 progenitors expressing the *Nkx2.2* gene produces visceral motor neurons, such as FMN and IEE precursors, which will differentiate into MOC and LOC efferent neurons [49,50]. To confirm that *Hoxb1* was specifically downregulated in the r4 ventral motor domain of the *CKONkx2.2* line, *in situ* hybridization on embryological (E) 11.5-old flat-mounted hindbrains confirmed selective loss of *Hoxb1* transcript in ventral r4 (**Fig 4A**). Moreover, *in situ* hybridization of *Tbx20*, a known marker for visceral motor neurons along the AP axis of the hindbrain [51–53], on coronal sections of postnatal (P) 8 pups showed a decreased population of Tbx20-expressing neurons in the ventral region of *CKONkx2.2* and *CKOr4* brains when compared to *Ctrl* brains (arrowheads in **Fig 4B**). No OC neurons were labeled in the expected ventral locations in *null* embryos since *Hoxb1* is inactivated from the onset of its expression before efferent neuron development (**Fig 4B**). Moreover, abnormally dorsally located cells expressing Tbx20 were also visible in all mutant brains (**Fig 4B**). These ectopic cells might be r3-like misspecified motor neurons, in line with the r4 to r3 change of identity upon loss of *Hoxb1* during early development [25,28,36].

To functionally assess auditory performances, we probed the activation of the auditory system in the *Hoxb1* conditional *CKONkx2.2* and *CKOr4* mutant mice and compared them with the *null* group (**Fig 2F–2H**). As shown in our earlier study [28] and in **Figs 1 and 2**, the constitutive inactivation of *Hoxb1* leads to threshold shifts at all the probed frequencies (mean threshold: 68.1 ± 1.8 dB SPL in **Fig 2F and S1 Data**). In the *CKOr4* mouse line, we measured an increase in threshold compared to *Ctrl* mice (mean threshold: 33.9 ± 0.8 dB SPL vs 63.2 ± 2.2 dB SPL in *Ctrl* mice and *CKOr4* mice respectively, $P = 2.1.10^{-8}$, two-tailed Mann-Whitney Wilcoxon's test, **Fig 2G and S1 Data**). In the same manner, we measured a significant elevation in threshold in the *CKONkx2.2* line compared to *Ctrl* mice (mean threshold: 33.9 ± 0.8 dB SPL vs 49.7 ± 3.3 dB SPL in *Ctrl* and *CKONkx2.2* mice respectively, *P = 0.0002*, two-tailed Mann-Whitney Wilcoxon's test, **Fig 2H and S1 Data**), thus partially reproducing the severe *null* auditory impairment. By testing the *Cre-recombinase* lines alone, which showed no difference compared to *Ctrl*, we were able to exclude any contribution of *r4-Cre* and *Nkx2.2-Cre* to the altered threshold of conditional mice (**S1 Fig**).

However, we noticed that the audiograms of the *CKONkx2.2* and *CKOr4* mice were widely distributed in comparison to the other genotypes. Thus, to determine the degree of variability

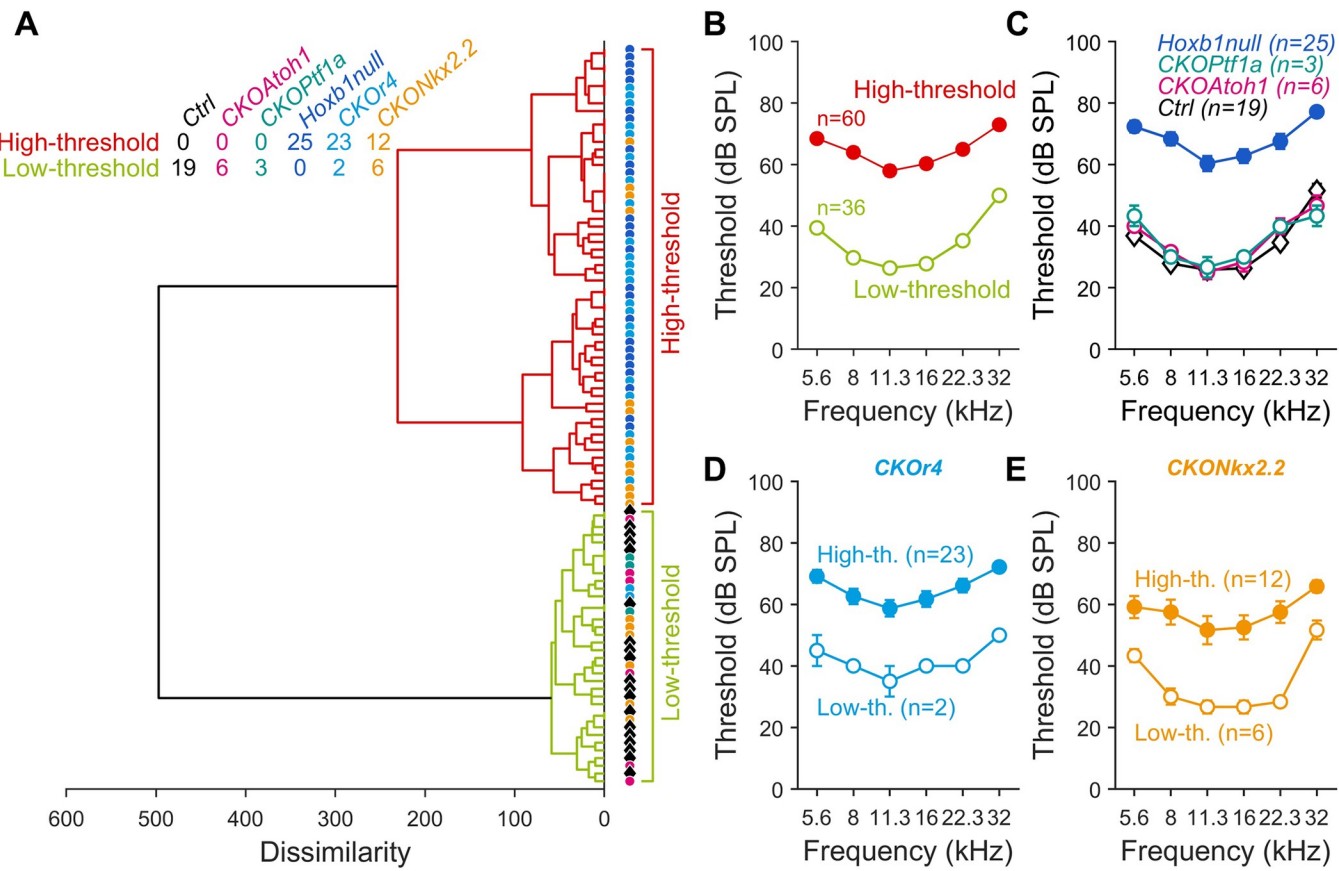

**Fig 5. Audiograms of low- and high-threshold conditional *Hoxb1* mutant mice.** (A) Ascending hierarchical classification (euclidean distance, ward linkage method) across the audiogram waveforms of *Ctrl*, *CKOAtoh1*, *CKOPtf1a*, *Hoxb1null*, *CKOr4* and *CKONkx2.2* mice. Cross tabulation inset shows the number of animals belonging to low- and high-threshold groups. **B)** Audiograms from all mice according to the low- and high-threshold distribution. (**C-E**) Audiograms from mice according to the low- and high-threshold distribution in (**C**) *Ctrl*, *CKOAtoh1*, *CKOPtf1a* and *Hoxb1null* mice, (**D**) *CKOr4* and (**E**) *CKONkx2.2* mice. Empty circles: low-threshold cluster, filled circles: high-threshold cluster. n indicates the number of cochleae recorded. See also **S1** Data.

in the hearing impairment, we used an algorithm of ascending hierarchical classification across the audiogram waveforms of the different mouse lines (96 audiograms, 6 lines, **Fig 5**). Two main classes of audiograms came out from the analysis, corresponding to mice with low- and high-threshold, respectively (see dendrogram in **Fig 5A**; cross-tabulation as shown in **Fig 5A**, $\chi^2 = 71.8$, $p = 6.1.10^{-14}$, low- and high-threshold audiograms in **Fig 5B and S1 Data**). Notably, audiograms of *Ctrl*, *CKOAtoh1*, and *CKOPtf1a* mice turned out to be part of the low-threshold class, according to their similar audiogram waveform (**Fig 5C and S1 Data**). On the opposite, all the *Hoxb1null* mice were segregated within the high-threshold group (**Fig 5C and S1 Data**). Among the *CKOr4* mice line, 92% of the animals were classified as high-threshold leaving only a small fraction of *CKOr4* mice in the low-threshold pool (**Fig 5A and 5D and S1 Data**). Moreover, 67% of the *CKONkx2.2* mice fell into the high-threshold category while 33% belonged to the low-threshold class (**Fig 5A and 5E and S1 Data**). This subdivision into high- and low-threshold groups can be explained by the efficiency and/or temporal variability of the action of the *Cre recombinase* during motor neuron development, as previously described for the *CKOr4* line [28]. In the mouse mutants with low auditory threshold, the *Cre* inactivates *Hoxb1* either inefficiently or after the specification of a suitable number of efferent neurons, so that OHC function is supported. Instead, the *Cre* would act with major efficiency or earlier, thus strongly affecting the efferent neuron formation in mice with a high auditory threshold.

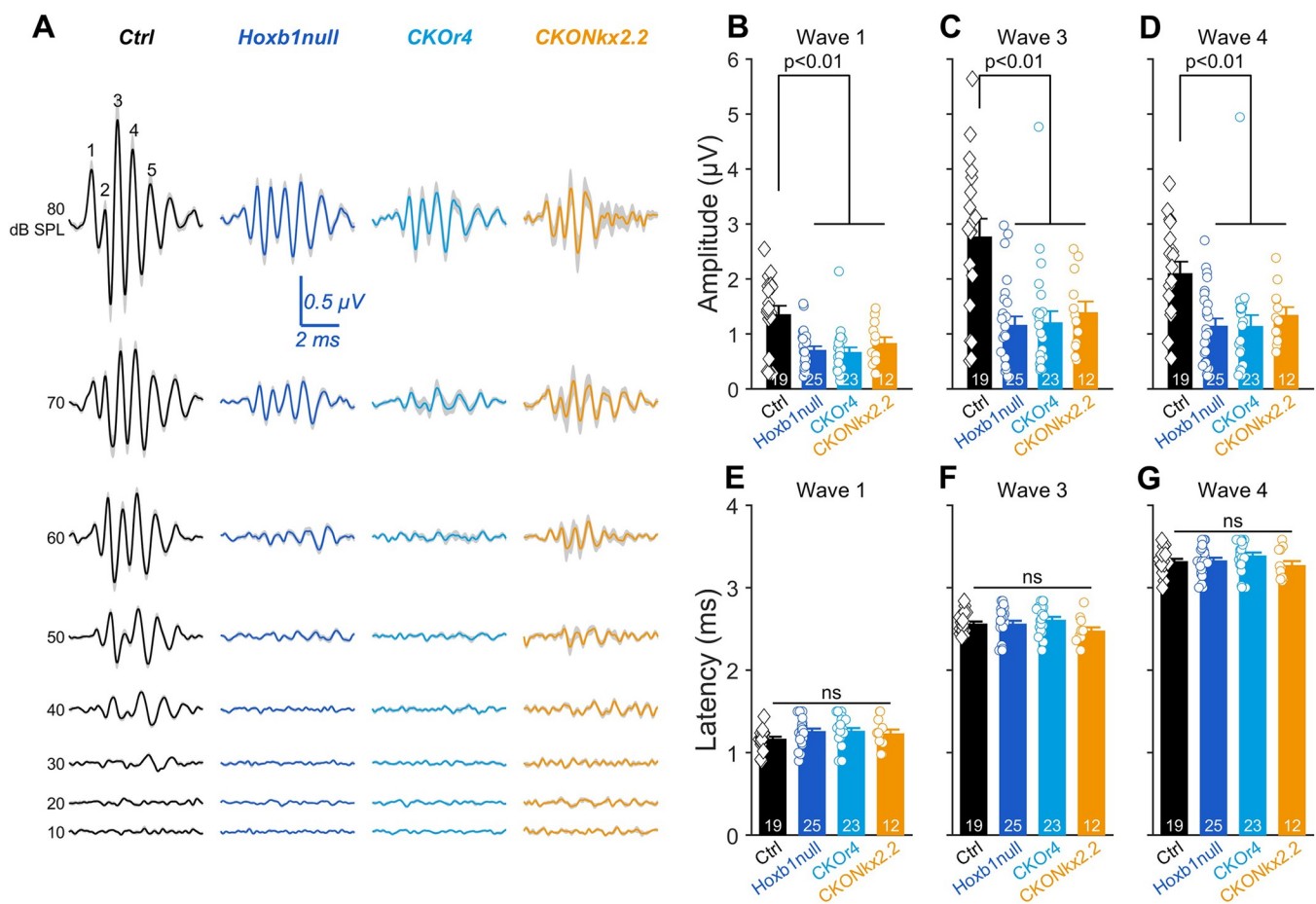

**Fig 6. Reduction of auditory nerve activity in r4-motor *Hoxb1* conditional mice.** (**A**) Mean ABR recordings evoked by 11.3-kHz tone burst from 10 to 80 dB SPL in *Ctrl*, *Hoxb1null*, *CKOr4* and *CKONkx2.2* mice. SEM: shaded grey. (**B-G**) Mean ABR wave 1, 3 and 4 amplitudes (B, C and D) and latencies (E, F and G) evoked at 11.3 kHz, 80 dB SPL in *Ctrl*, *Hoxb1null*, *CKOr4* and *CKONkx2.2* mice (the genotypes refer to the same colors in A). Symbols: individual cochlea. Samples sizes are indicated by numbers. See also **S1** Data.

Finally, to probe the functional alteration of the ascending auditory pathway, we measured the amplitude and latency of the most prominent ABR waves, i.e., wave 1, 3 and 4, corresponding to the synchronous response of neurons in the auditory nerve, the superior olivary complex and the lateral lemniscus, respectively (**Fig 6**). Consistent with the elevated thresholds (**Fig 6A and S1 Data**), we measured smaller ABR wave 1, 3 and 4 amplitudes in the *null* group (Amp$_{wave1}$: 0.71 ± 0.07 μV, Amp$_{wave3}$: 1.16 ± 0.16 μV; Amp$_{wave4}$: 1.15 ± 0.14 μV), but also in *CKOr4* (Amp$_{wave1}$: 0.67 ± 0.09 μV, Amp$_{wave3}$: 1.2 ± 0.21 μV; Amp$_{wave4}$: 1.14 ± 0.2 μV) and *CKONkx2.2* mice (Amp$_{wave1}$: 0.83 ± 0.11 μV, Amp$_{wave3}$: 1.39 ± 0.20 μV; Amp$_{wave4}$: 1.34 ± 0.15 μV) belonging to the high-threshold group in comparison to *Ctrl* mice (Amp$_{wave1}$: 1.35 ± 0.16 μV, Amp$_{wave3}$: 2.77 ± 0.33 μV; Amp$_{wave4}$: 2.10 ±0.22 μV; wave 1: P < 0.05, wave 3: P < 0.01, wave 4: P < 0.05, one-way analysis of variance and *post hoc* Tukey test) (**Fig 6B–6D and S1 Data**). In contrast, latencies were comparable between the genotypes (**Fig 6E–6G and S1 Data**) consistent with our previous results [28]. Taking together, these results suggest that the selective invalidation of *Hoxb1* in r4 ventral motor neurons (*CKONkx2.2*) leads to hearing impairments that partially reproduce the ones observed in the *null* and *CKOr4* for the experimental groups with high thresholds.

## Altered MOC innervation and OHC morphology in r4-motor conditional *Hoxb1* mutants with abnormally high thresholds

To assess whether the heterogeneity in the auditory thresholds was correlated with structural defects at the level of the inner ear, we processed the previously recorded cochleae for EM analyses (**Fig 7**). In *Ctrl* mice, proper MOC endings at the level of OHCs and their characteristic arrangement in several rows could be observed (**Fig 7A and 7B**). Similarly, the low-threshold groups of *CKOr4* and *CKONkx2.2* mouse mutants with normal auditory thresholds displayed MOC efferent nerve endings on OHCs and V-shaped stereocilia with staircase pattern, similarly to *Ctrl* inner ears (**Fig 7E, 7F, 7I and 7J**; *CKONkx2.2low vs Ctrl: 88.7 ± 7.7%; P = 0.5; CKOr4low vs Ctrl: 78.6± 8.6%; P = 0.2;* **S2 Fig** and **S1 Data**), suggesting that presence of MOC innervation could preserve normal OHC morphology. On the contrary, in the high-threshold groups of *CKOr4* and *CKONkx2.2* mouse mutants with elevated auditory thresholds, no or very few efferent endings could be found at the level of the OHCs in all cochleae examined (**Fig 7G and 7K**; *CKONkx2.2high vs Ctrl: 0.7 ± 0.1%; P<0.0001; CKOr4high vs Ctrl: 0 ± 0%; P<0.0001;* **S2 Fig** and **S1 Data**), and OHC stereocilia showed severe structural abnormalities with affected cilia either fused or absent (**Fig 7H and 7L**) highly reproducing the *null* phenotype (**Fig 7C and 7D**) [28]. These cellular abnormalities support the functional auditory impairments and suggest that the specific absence of motor efferent neuron innervations is strictly linked to the malformation of OHCs and is, most probably, responsible, even if not completely, for the altered auditory threshold in *Hoxb1*-dependent hearing loss.

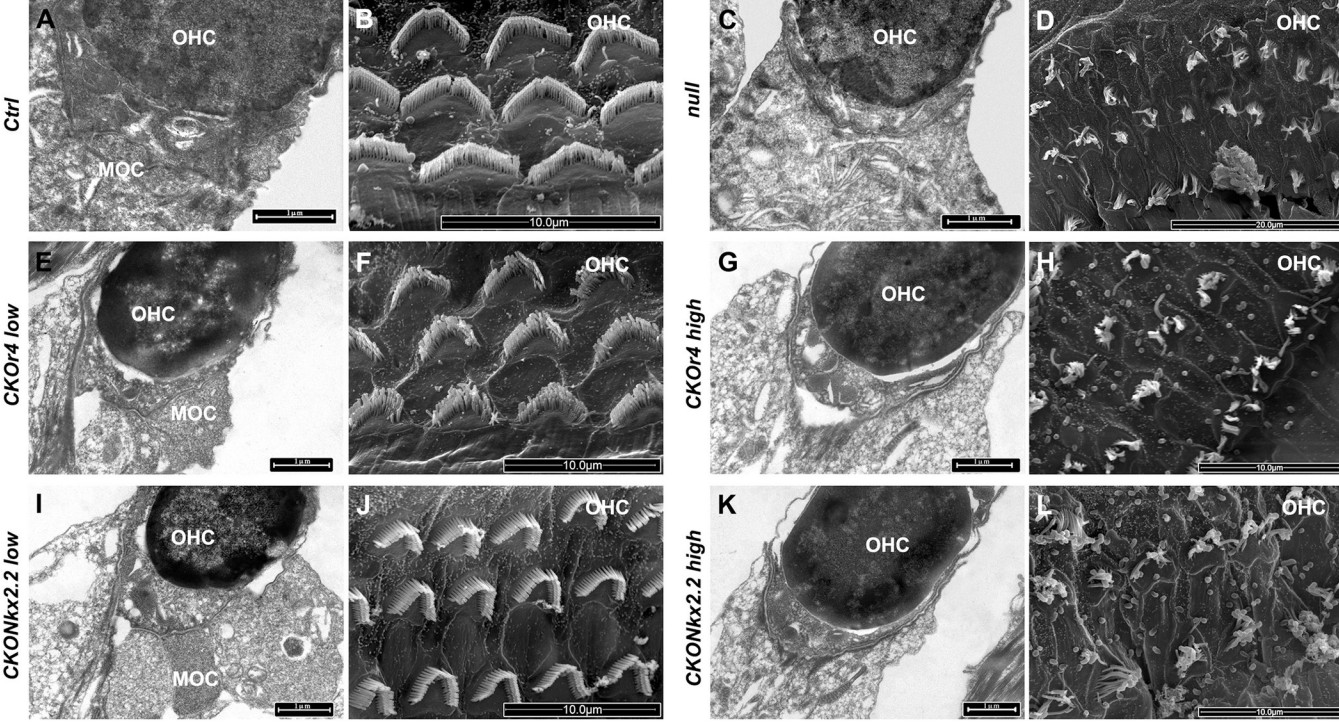

**Fig 7. Altered medial olivocochlear neuron (MOC) innervation and outer hair cell (OHC) morphology in r4-motor conditional *Hoxb1* mutants.**
Transmission electron microscopy (TEM) of MOC innervating OHC (**A, C, E, G, I, K**) and scanning electron microscopy (SEM) of OHCs (**B, D, F, H, J, L**) in the cochleae of the genotypes indicated to the left. MOC innervation and normal OHC morphology were observed in control (*Ctrl*) and conditional mice with low auditory thresholds (A, B, E, F, I, J). In mutant mice with altered high thresholds (C, D, G, H, K, L), no MOC efferent on OHCs and severe malformations of OHC arrangement and disarray of stereocilia were observed. Scale bars 1μm (A, C, E, G, I, K), 10μm (B, D, F, H, J, L). See also S2 Fig and S1 Data.

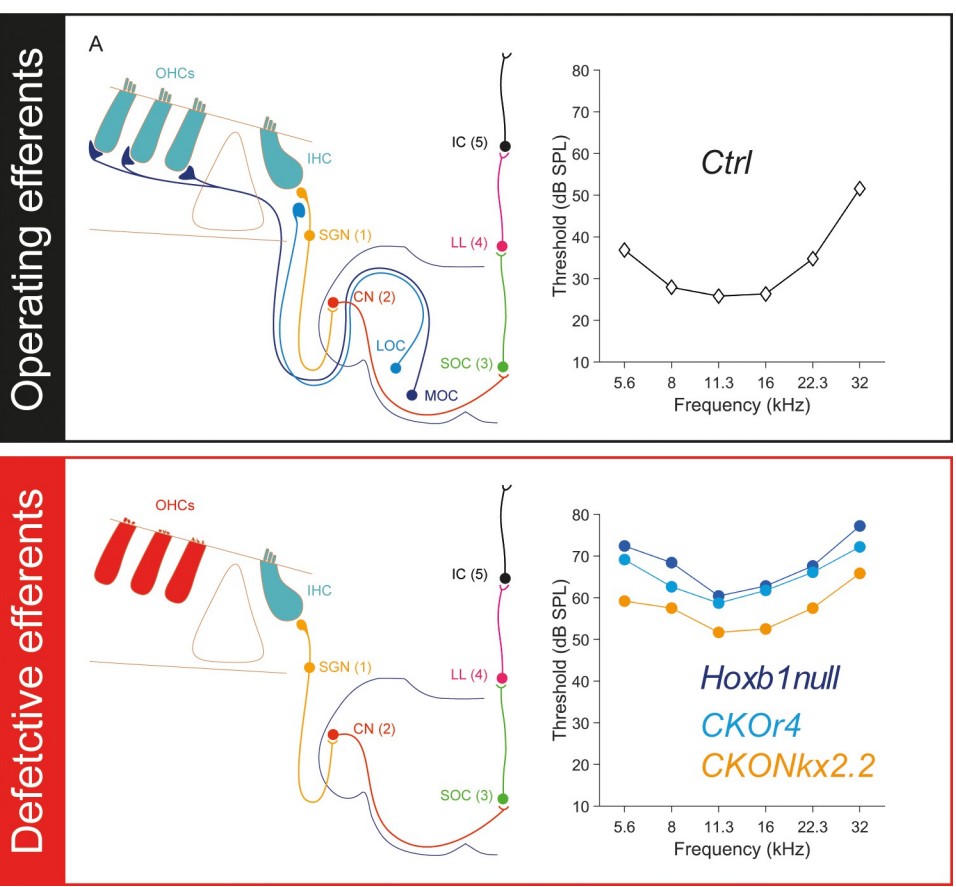

**Fig 8. Summary model of the effect of Hoxb1 inactivation in the auditory pathway.** In a physiologically normal condition in which efferent neurons are normally operating, the innervation of MOC and LOC efferent fibers to OHCs and to afferent fibers in contact with IHCs leads to a normal audiogram measured by ABR (to the right). In contrast, the presence of defective efferents, as is the case for full or conditional deletion of *Hoxb1*, will affect the normal shape of the audiograms causing increased auditory thresholds, as described in this study. Abbreviations: OHCs, outer hair cells; IHC, inner hair cell; MOC, medial olivocochlear neuron; LOC, lateral olivocochlear neuron; SGN, spiral ganglion neuron; CN, cochlear nucleus; SOC, superior olivary complex; LL, lateral lemniscus; IC, inferior colliculus.

## Discussion

### Development and function of efferent innervation in the auditory pathway

Our aptitude to successfully perceive and interact with the environment includes the activity of efferent motor pathways that can modulate the signals transmitted by the afferent sensory systems. In the auditory pathway, the efferent system, constituted by the OC bundle and originating from the medial and lateral OC nuclei in the superior olivary complex of the brainstem, projects onto distinct hair cells of the developing cochlea and is well recognized to modulate cochlear activity [54]. MOC and LOC efferents play distinct roles in regulating the cochlear function [5,55]. While MOC terminals use principally acetylcholine to inhibit OHCs and modulate sound amplification within the cochlea [13], LOC efferent fibers contain several neurotransmitters, such as dopamine, acetylcholine, and GABA, project directly onto afferent fibers and regulate auditory fibers activity [56–58], and binaural balance for spatial sound localization [59].

Auditory processing deficits can occur in models of congenital OC dysfunction, even if these deficits have often been suggested to reflect abnormal central auditory development

rather than defects in the OC feedback [60]. Direct evidence that the OC system contributes to the perception of sounds in noise comes primarily from animal studies in which the OC bundle was surgically ablated. Lesions of the crossed OC bundle in adult cats result in small or no increases in tone detection thresholds and intensity discrimination [61–63] although de-efferentation renders the cochlea more vulnerable to acoustic injury [64] and dramatically increases noise-induced permanent threshold shifts [23]. However, severed OC bundle in neonatal cats leads to elevated thresholds, decreased sharpness of tuning, as well as lower spontaneous discharge rates [65], most probably because OHCs, as cochlear amplifiers, fail to develop normally in the absence of efferent innervation. This also highlights the importance of early interactions, possibly of chemical nature, between efferent neurons and their postsynaptic targets more at neonatal than adult stages [3] for proper cochlear functional maturation.

These studies, however, rely on transection of the entire OC bundle, including both LOC and MOC neurons, and are thus unable to distinguish MOC from LOC function. Unilateral chemical ablation of LOC neurons *via* neurotoxin injection into the lateral superior olive complex disrupts the normal interaural correlation in response to sounds of equal intensity by having, however, no effect on auditory thresholds but rather on their amplitude [7,59]. Indeed, increases in suprathreshold neural responses turned out to be frequency- and level-independent and not attributable to OHC-based effects [7]. These findings suggest that LOC neurons modulate cochlear nerve excitability and protect the cochlea from neural damage, but have little or no effect on auditory thresholds [56].

## Impact of efferent motor neurons on the auditory threshold impairment of *Hoxb1null* mutants

Thanks to a cell-specific conditional genetic approach in mice, we show that severe reduction or loss of early IEE generation during development will lead to hearing defects and severe abnormalities of OHC morphology (**Fig 8**). Our earlier and present studies have demonstrated a strong auditory impairment in *Hoxb1null* mice as confirmed by ABR and DPOAE (**Fig 1** and [28]). Since r4-derived sensory and motor auditory components are both affected in these constitutive *null* mice, we were unable to discern the implication of sensory cochlear *versus* motor efferent neurons in this defect. We thus used a conditional genetic approach allowing us to independently inactivate *Hoxb1* either in the r4-derived cochlear sensory neurons or in the efferent motor neurons and compared these mice to *null* and *CKOr4* mutant mice in which both types of neurons were altered [28]. Our work shows that altered auditory threshold and malformation of OHCs are partially reproduced in *CKONkx2.2* conditional mice in which *Hoxb1* is solely inactivated in ventral motor neurons at early stages during the onset of IEE generation. These results argue for a key role of efferent motor neurons in the impairment of the cochlear amplification system in the absence of Hoxb1 function.

A direct role of *Hoxb1* on hair cell development can be excluded since the gene is not expressed in presumptive hair cells [28]. In addition, altered hearing thresholds are already present in one-month-old *Hoxb1null* mice acoustically isolated at birth, suggesting that the increased threshold is probably not due to altered function of the efferent feedback systems that cannot protect the cochlea from noise-induced hearing damage [28]. Moreover, our previous data showed that the morphology of OHCs was only affected postnatally after MOC neurons and OHCs failed to interact, whereas OHCs resulted undamaged at P8 before interactions normally occur [28]. Moreover, in neonatal transection of the OC bundle, MOCs never reach the OHCs in the de-efferented ears [65]. This strongly suggests that MOC neurons might either over-or under-express a key component of the OHC amplification process. More support comes from the observations that the persistence of MOC neurons innervating the

OHCs in the low-threshold groups of *CKOr4* and *CKONkx2*.2 mice (this study) or in the partially de-efferented cats [65] is sufficient to support a normal auditory threshold and OHC morphology. In addition, no differences in the latencies of the evoked responses were found in the ABR analysis of the mutant mice, indicating that the auditory stimuli, when perceived, can travel normally along the successive nuclei of the central auditory pathway, even in the presence of abnormal CN and ventral lateral lemniscus specification, as previously described [28]. However, the amplitude of the different waves resulted impaired in mutant mice with high thresholds, suggesting that the absence of LOC neurons might also contributes to the hearing defect, particularly in regulating amplitude levels, as shown by chemical LOC disruptions [7].

We, therefore, propose that degeneration of OHCs and consequently, altered hearing thresholds in our mutant mice, might be caused by the absence of synaptic and/or trophic chemical stimulation of cochlear hair cells from the MOC fibers during a postnatal critical period, which is essential for proper maturation and functioning of OHCs. Nevertheless, we have to consider that the altered function of the MOC reflex (due to loss of MOC neurons and misspecification of PVCN nucleus) and of the MEM reflex (due to affected differentiation of FMN neurons) [28,39], even if, most probably, not the main cause of the altered threshold, render the cochleae of *Hoxb1* mutants more hypersensitive to environmental sounds. Therefore, the OHCs (and consequently the cochlear amplification system) will result increasingly damaged with time determining a more pronounced progressive age-related degeneration of hearing in *Hoxb1* mutant mice, as previously described [28]. Finally, in order to exclude as much as possible the influence of the MEM reflex, which mainly responds to high-level sound stimuli [16–18], we have chosen to use low-level sound stimuli when measuring contralateral suppression.

In conclusion, we have dissected the specific functional role of r4-derived efferent motor neurons in the complex phenotype of the *Hoxb1null* mutant. Our present work has provided strong evidence in favor of the important role of early MOC efferent innervation on the functional maturation of OHCs unraveling one of the causes of altered auditory threshold in the sensorineural hearing loss of mice and patients with a recessive mutation in the *HOXB1* gene.

## Materials and methods

### Ethics statement

All animal experiments were conducted in accordance with the French Animal Welfare Act and European guidelines for the use of experimental animals, using protocols approved by the French Ministry of Education, Research and Innovation (reference # APAFIS#l 8019–2018112919027679 v4) and the local ethics committee in Nice, France (CIEPAL NCE/2019–548).

### *Hoxb1* transgenic mutant lines

The generation and genotyping of mouse lines have been previously described: *Hoxb1flox*, *Hoxb1null*, and *r4-Cre* line, before denominated *b1r4-Cre* [28], *Nkx2.2-Cre* [48], *Atoh1-Cre* [42] and *Ptf1a-Cre* [44]. To alter Hoxb1 function in r4 motor or sensory neurons, *Hoxb1flox* mice were crossed with several knock-in DV-*Cre recombinase* lines: *Nkx2.2-Cre* for motor neurons, *Atoh1-Cre* and *Ptf1a-Cre* for glutamatergic and GABAergic/glycinergic cochlear neurons respectively, obtaining conditional KO line of specific DV domains, *CKONkx2.2*, *CKOAtoh1* and *CKOPtf1a*. Instead, in the *r4-Cre* line the *Cre recombinase* is under the control of a well-characterized enhancer that induces the later expression of *Hoxb1* exclusively in r4 [66]. The *r4-Cre* line were crossed with *Hoxb1flox* mice to inactivate *Hoxb1* in r4 motor and sensory neurons from E8.5 onwards obtaining conditional KO of r4, *CKOr4*, previously denominated

*Hoxb1*[lateCKO] [28]. Mice were bred in-house and maintained on a 129S2/SvPas genetic background. Female and male mice were put in mating in the evening, and midday of the day of the observed vaginal plug was considered as embryonic day 0.5 (E0.5).

## Distortion Product Otoacoustic emissions (DPOAE)

An ER-10C S/N 2528 probe (Etymotic Research), consisting of two emitters and one microphone, was inserted in the left external auditory canal. Stimuli were two equilevel (65 dB SPL) primary tones of frequency f1 and f2 with a constant f2/f1 ratio of 1.2. The distortion 2f1-f2 was extracted from the ear canal sound pressure and processed by HearID auditory diagnostic system (Mimosa Acoustic) on a computer (Hewlett Packard). The probe was self-calibrated for the two stimulating tones before each recording. f1 and f2 were presented simultaneously, sweeping f2 from 20 kHz to 2 kHz by quarter octave steps. For each frequency, the distortion product 2f1-f2 and the neighboring noise amplitude levels were measured and expressed as a function of f2.

## Auditory brainstem response (ABR)

Mice were anesthetized by intraperitoneal injection of Rompun 2% (3 mg/kg) and Zoletil 50 (40 mg/kg) and mouse temperature was measured with a rectal thermistor probe and maintained at 38.5˚C ± 1 using a heated under-blanket (Homeothermic Blanket Systems, Harvard Apparatus). The acoustical stimuli consisted of 10-ms tone bursts, with an 8-ms plateau and 1-ms rise/fall time, delivered at a rate of 20.4/s with alternate polarity by a JBL 2426H loudspeaker in a calibrated free field. Stimuli were generated and data acquired using MATLAB (MathWorks, Natick, MA, USA) and LabVIEW (National Instruments, Austin, TX, USA) software. The potential difference between vertex and mastoid intradermal needles was amplified (5000 times, VIP-20 amplifier), sampled (at a rate of 50 kHz), filtered (bandwidth of 0.3–3 kHz) and averaged (600 times). Data were displayed using LabVIEW software and stored on a computer (Dell T7400). The ABR was recorded at decreasing intensity of sound from 100 to 10 dB SPL and ABR threshold was defined as the lowest sound intensity, which elicits a clearly distinguishable response. The ABR thresholds were measured at 6 test frequencies 5.6, 8, 11.3, 16, 22.3, 32 kHz that are preferentially perceived at different levels of cochlea across the tonotopic axis, and each audiogram was stored as a vector containing these six threshold values. Audiograms obtained in control and mutant mice were classified using an agglomerative hierarchical clustering algorithm using Euclidean distance and Ward linkage method. Unless specified, Shapiro-Wilk test (p > 0.05) and one-way ANOVA test with *post-hoc* Tukey's multiple comparisons test were used for statistical tests.

## Scanning and transmission electron microscopy

The cochleae of *WT*, *null*, and of the different conditional *KO* lines, previously tested by ABR, were dissected to analyze the efferent innervation and OHC morphology. One cochlea of each animal was analyzed by transmission electron microscopy (TEM) and the other by scanning electron microscopy (SEM), as previously described [28]. The inner ear was fixed in 2.5% glutaraldehyde in 0.1 M phosphate-buffered saline (PBS) pH 7.4 for 4 h at 4˚C and rinsed in PBS overnight. Corti organs were isolated, rinsed in PBS and postfixed in 1% osmium tetroxide solution in a 0.05 M PBS at pH 7.4. Samples for TEM were dehydrated with ethanol and then propylene oxide and embedded in Epon 812 resin (Fluka). The blocks were cut using a Super Nova Leica Ultratome. Ultrathin sections (80 nm) were stained with 2% uranyl acetate and 2.66% lead citrate. Grids were examined by using a Fei-Tecnai-G electron microscope at 120 kV. For SEM analysis, after fixation and dehydration in alcohol, a critical point drying was

performed. The samples were mounted on aluminum stubs and sputter coated with gold. The processed samples were investigated and photographed using a JEOL 6700F SEM operated at 5 kV and at 8.3 mm working distance. SEM and TEM images were collected digitally. We processed n = 3 *Ctrl*, n = 6 *CKONkx2.2*, n = 6 *CKOr4*, n = 3 *null* cochleae.

## *In situ* hybridization

*In situ* hybridization on brain sections and whole mount embryos were performed as previously described [28]. P8 pups were perfused with 4% paraformaldehyde (PFA) in phosphate-buffered saline (PBS) pH 7.4. The brains were dissected and fixed overnight in 4% PFA. Tissues were cryoprotected with 10, 20 and 30% sucrose in PBS, frozen in OCT embedding matrix (Kaltek) and cryostat sectioned at 16 μm thickness in the coronal plane. The coronal brain sections were hybridized using digoxigenin-labelled (Roche labelling kit) riboprobes for *Tbx20* and acquired at slide scanner Vectra Polaris. Whole mount hybridization was performed on E11.5 embryos after fixation overnight in 4% PFA in PBS using the antisense RNA probe for *Hoxb1*. Hindbrains were dissected and flat mounted in 4% PFA:80% glycerol and acquired at stereomicroscope.

## Supporting information

**S1 Fig. Normal Auditory Brainstem Response in *DV-Cre* and *r4-Cre recombinase* lines.** Audiograms from *r4-Cre*, *Nkx2.2-Cre*, *Atoh1-Cre*, *Ptf1a-Cre* mice. Auditory thresholds in *Cre recombinase* lines are comparable to *Ctrl* mice. Mean threshold ± SEM: 33.9 ± 0.8 dB SPL, 38.3 ± 1.4 dB SPL, 35 ± 2.5 dB SPL, 34.6 ± 0.8 dB SPL, 32.9 ± 2.3 dB SPL in *Ctrl*, *r4-Cre*, *Nkx2.2-Cre*, *Atoh1-Cre*, *Ptf1a-Cre* mice respectively. P = 0.27, one-way analysis of variance. See also **S1** Data.
(TIF)

**S2 Fig. Altered number of MOC/OHC interactions in mutant cochleae with high auditory thresholds.** The percentage of the presence (**A**) of MOC/OHC interactions was quantified relative to the *control* group (*CKONkx2.2low* vs *Ctrl*: 88.7 ± 7.7%; P = 0.5; *CKOr4low* vs *Ctrl*: 78.6 ± 8.6%; P = 0.2; *CKONkx2.2high* vs *Ctrl*: 0.7 ± 0.1%; P<0.0001; *CKOr4high* vs *Ctrl*: 0 ± 0%; P<0.0001), whereas the percentage of the absence (**B**) of MOC/OHC interactions was quantified relative to the *Null* group (*CKONkx2.2low* vs *Null*: 13.0 ± 6.8%; P<0.0001; *CKOr4low* vs *Null*: 21.5 ± 7.6%; P<0.0001; *CKONkx2.2high* vs *Null*: 100 ± 6.8%; P>0.9; *CKOr4high* vs *Null*: 100 ± 6.8%; P>0.9). Data were statistically analyzed with One-Way ANOVA followed by the Bonferroni test and results are shown as mean ± SEM.**** P<0.0001. See also **S1** Data.
(TIF)

**S1 Data. Numerical raw data related to Fig 1C–1F, Fig 2C–2H, Fig 5A–5E, Fig 6A–6G, S1 Fig, S2 Fig.**
(XLSX)

## Acknowledgments

We thank L. Gan for the *Atoh1-Cre* line and Christopher V. E. Wright for the *Ptf1a-Cre* line that we kindly received from Huda Y. Zoghbi, as well as A. Holz for the *Nkx2.2-Cre* line. A particular thank to K. Moneret and M. Giribaldi for animal husbandry.

## Author Contributions

**Conceptualization:** Maria Di Bonito, Bice Avallone, Regis Nouvian, Michèle Studer.

**Data curation:** Maria Di Bonito, Jérôme Bourien, Bice Avallone, Regis Nouvian, Michèle Studer.

**Formal analysis:** Maria Di Bonito, Jérôme Bourien, Monica Tizzano, Anne-Gabrielle Harrus, Bice Avallone, Regis Nouvian.

**Funding acquisition:** Jean-Luc Puel, Michèle Studer.

**Investigation:** Maria Di Bonito, Monica Tizzano, Anne-Gabrielle Harrus, Jean-Luc Puel, Michèle Studer.

**Methodology:** Maria Di Bonito, Monica Tizzano, Anne-Gabrielle Harrus, Bice Avallone, Regis Nouvian.

**Project administration:** Jean-Luc Puel, Regis Nouvian, Michèle Studer.

**Resources:** Jean-Luc Puel, Bice Avallone, Regis Nouvian, Michèle Studer.

**Software:** Jérôme Bourien.

**Supervision:** Bice Avallone, Regis Nouvian, Michèle Studer.

**Validation:** Maria Di Bonito, Jérôme Bourien, Monica Tizzano, Bice Avallone, Regis Nouvian, Michèle Studer.

**Visualization:** Maria Di Bonito, Bice Avallone, Regis Nouvian, Michèle Studer.

**Writing – original draft:** Maria Di Bonito, Michèle Studer.

**Writing – review & editing:** Maria Di Bonito, Jérôme Bourien, Jean-Luc Puel, Bice Avallone, Regis Nouvian, Michèle Studer.

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
