## [Decision Letter · Decision Letter 0]

30 May 2023

Dear Dr Studer,

Thank you very much for submitting your Research Article entitled 'Abnormal outer hair cell efferent innervation affects Hoxb1-dependent sensorineural hearing loss' to PLOS Genetics. Due to the challenge of identifying competent and available reviewers, we apologize for the delay in reaching an editorial decision.

Your article has been evaluated by the journal's senior editors and by three independent peer reviewers. The study, based on the use of multiple mouse lines with constitutive and conditional knockouts of Hox1 genes, is a real tour de force of mouse genetics. The approach is sound and achieves the disruption of different parts of the auditory signaling pathways to dissect the networks involved in hearing loss in Hoxb1 mouse mutants. The three reviewers agree that this research is important and the manuscript is well written. 

The Reviewers also agree that while the present study extends previous work conducted by your group, the current use of conditional mouse mutagenesis leads to define the cellular basis of Hoxb-1-linked hearing loss. The Reviewers concur that major strengths of this study include: (a) the use and characterization of four conditional mouse mutant lines, (b) the thorough electrophysiological analysis and high-resolution anatomical methods that have been employed, (c) the side-by-side comparison of conditional and null Hoxb1 mutant mice, and (d) the analysis of audiograms in all mouse mutant lines. As you will see, there is an overall consensus regarding the potential interest and significance of your work.

However, a number of weaknesses were also identified, as described below. All three Reviewers raised concerns that revolve around the lack of details regarding the afferent and efferent fibers based on the use of tracing and/or immunocytochemistry experiments, which were not conducted; the need for a deeper characterization of the conditional mouse mutant lines; the requirement for additional marker analysis; the need for better quantification of the data; and the lack of appropriate citations of previous papers that are of high relevance to the current study, including the omission of recent literature from the references. As a result, all the Reviewers expressed concerns that not all conclusions that were reached are supported by the data presented in the current manuscript.

Specifically, Reviewer 1 raises serious concerns regarding the lack of mapping of the inner ear efferents (IEE) fibers with proper tracing to the ear in the relevant cKO mice. The Reviewer also requests to add the statistical significance to specific figures; to provide more robust proof of the absence of medial olivocochlear (MOC) neuron efferents by using a dye tracing, and to perform immunostainings for type II afferents.

Reviewer 2 states that there is not enough evidence to demonstrate that MOC neuron efferent terminals are consistently aberrant in the cKO animals and that the authors do not convincingly demonstrate whether the defects observed in the knockout mice are limited to MOCs, or if there are also contributing perturbations caused by the middle ear muscles. This Reviewer is concerned that contributions from other systems may indeed have an impact on the observed phenotypes. Therefore, the Authors should at least discuss the potential contributions of other systems to the reported pathology in the cKO mice. Reviewer 2 also requires more robust evidence of MOC axon terminal degeneration, stating that at present only one single animal is used for each condition, with only one single image shown and little detail regarding numbers. This is not considered sufficient evidence. 

Reviewer 3 recommends detailed pre-synaptic and post-synaptic marker analysis, which would significantly strengthen the conclusion that interactions between motor neurons and outer hair cells are crucial. Regrettably, such marker analysis is lacking. This Reviewer also requests further support to the conclusion reached by the Authors that, based on analysis of Tbx20, lateral olivocochlear (LOC) and MOC neuron populations are decreased. The Reviewer states that it remains unclear whether LOC and MOC neurons are normally born but fail to differentiate and extend axons and dendrites, or if they are not born at all, or if they are born and then die. Experiments with additional MN markers (GATA3, ChAT) or cell death markers would help distinguish between these possibilities. Like Reviewer 1, Reviewer 3 also requests the Authors add the statistical significance and error bars to specific figures.

The article and the critiques from the three Reviewers have now been discussed among members of the editorial board with appropriate expertise. At present, we all agree that the work required to address the Reviewers' (and our) concerns requires a ‘Major Revision’. If additional information is obtained in response to all of the critiques raised by the Reviewers, we will be willing to evaluate a revised manuscript that incorporates all of the revisions that have been requested. However, we caution that eventual success at PLOS Genetics will require satisfactorily addressing all of the main concerns that were raised by the Reviewers.

If you decide to revise the manuscript for further consideration at PLOS Genetics, please aim to resubmit within the next 60 days, unless it will take extra time to address the concerns of the reviewers, in which case we would appreciate an expected resubmission date by email to plosgenetics@plos.org.

To resubmit, use the link below and 'Revise Submission' in the 'Submissions Needing Revision' folder. Please do not hesitate to contact us if you have any concerns or questions.

Yours sincerely,

Licia Selleri

Academic Editor

PLOS Genetics

Hua Tang

Section Editor

PLOS Genetics

Reviewer's Responses to Questions

**Comments to the Authors:**

Reviewer #1: This paper presents the effects of Hoxb1 that can affects IEE. An altered effects of Nkx2.2 CKO after deletion of IEE and FBMs while losing with Hoxb1 using Atoh1 and Ptf1a CKO shows near normal cochlear nuclei.

Unfortunately, the data of IEE to vestibular efferents requires additional information. Given that the IEE is somewhat bilateral to the MOC the statement cited in #48 is an overstatement after cutting the OC bundle (see details of Bruce et al., 1997).

Previous work showed that the loss of OHC and IHC after losing it afferents and efferents (Kersigo and Fritzsch, 2015). I strongly add the details of afferents to OHC using peripherin in P8 and adult mice (Elliott et al., 2021). I suggest tracing the IEE fibers with proper tracing to the ear.

Introduction’

P3, Lin 5 cochlear cranial nerves end up in the cochlear nuclei. The distinct population is clear but does not fit to the ‘vestibulocochlear’ nerves (see Yang et al., 2011; Filova et al., 2022). Suggest to refer to the cochlear cranial nerve to avoid misunderstandings.

P3, line 10 Suggest pointing out the facial MNS that migrate away (see Bruce et al., 1997).

P4, line 5 suggest adding r2-5 for auditory nuclei

P4, line 15 suggest citing the work of Wong et al., 2011 that details the r4 develoment.

P6, 9 from below, suggest giving credit to Chen et al., 2012 that details very nicely the absence of r4, thank you.

P8, line 11 from below. An additional IEE are known as vestibular efferents (see Bruce et al., 1997 for details). What will happen in the absence of IEE that goes to reach out the vestibular hair cells? Will that be a parallel next review?

P10 The absence of IEE is shown nicely with ISH. However, my question is that all IEE are all absent in the relevant CKO mice? Perhaps a tracing of efferents could validate the suggestion as we know that even in the absence of ear neurons that IEE can redirect to the facial MNs (Ma et al., 2000).

P11 line 16 why is a _ between cochlea and MOC?

References:

2 should be replaced by Malmierca, 2015

3 Simmons should be replaced by Simmons et al., 2011

15 I believe the paper by Zuo et al., 1999 should be cited here

19. suggest adding the citation of Elliott et al., 2021

28. suggest adding the work of Matei et al., 2005 for complete

30 a most recent presentation by Elliott et al., 2023

31 suggesting citations such as Bruce et al., 1997

34 suggest adding the work of Karis et al., 2000

55 suggest adding the citation of Wang et al., 2005

65 suggest adding for Wong et al., 2011 that details the feedback for Hox

Figures

Fig. 1 Significances are needed (p; 0.05?) in B. The 32 kHz shows an effect of Hoxb1 het. What is the significance?

Fig. 2 A recent publication (Elliott et al., 2023) shows the expansion of Ptf1a deletion. It would be interesting to show the effect of Ptf1 deletion in r4 that should show less expansion compared to Ptf1a deletions.

Fig. 4 A detailed tracing information was presented by Bruce et al., 1997 that clarifies the presentation of vestibular, LOC and MOCs. How many vestibular efferents develop in different CKO deletions?

Fig. 6 This information seems to compare with Maricich et al., 2009 after Hoxb1 deletions of Atoh1. Different ABRs have been described after complete deletion of Isl1 (Filova et al., 2023). How close is the ABR after different deletions?

Fig. 7. Nice images but will not proof the absence of MOCs. Suggest using a dye tracing to show the absence of MOCs (see Zuo et al., 1999).

Fig. 8 Suggest providing Prph type II fibers (see Elliot et al., 2021). Show the immunostaining for type II afferents and trace the LOC/MOC differentially at P8 and P40.

Reviewer #2: In the manuscript by Di Bonito et al (PGENETICS-D-23-00405) the authors use mouse lines with systemic and conditional knockouts of Hox1 genes (expressed in rhombomere 3-4 where auditory neurons originate), intended to disrupt different parts of the auditory signaling pathways to investigate the precise networks involved in hearing loss in Hoxb1 mouse mutants. The work combines auditory function testing including auditory brainstem response (ABR), distortion product otoacoustic emissions (DPOAE), scanning and transmission electron microscopy (EM), and in situ hybridization for markers of motor neurons to show that the cochlear innervation defects and hearing deficits observed in Hoxb1 null mice are likely due to defects specifically in the olivocochlear efferent pathway. Further, the results indicate that MOC innervation of the cochlea, and particularly outer hair cells (OHC), have a clear role in development and maintenance of the OHC system, which is critical for normal hearing but is aberrant in mice with mutations of the efferent system.

The manuscript is well written, figures are clear, statistics are appropriate. Some more recent literature is missing from citations (see detail below). Overall this is important work that furthers our knowledge of the role of the olivocochlear efferent system in maintaining cochlear cell health and hearing function by demonstrating that loss of olivocochlear neurons in developmental stages causes OHC degeneration.

Weaknesses of the work are that it depends on precise knockout of Hoxb1 in different cell populations using different Cre driver lines. This work does not demonstrate the precision of the knockouts, readers must reference previous works to see demonstrations of the specificity of the mouse lines. However, a helpful schematic is added to show how the cell-type specific knockouts are performed by targeting different rhombomeres in the developing nervous system. There are some variable results with the penetrance of the Cre in some of the conditional knockouts, resulting in some variable auditory function responses, but this is sufficiently discussed and analyzed in the manuscript. Another weakness is that there is not enough evidence to demonstrate that MOC efferent terminals are consistently aberrant in the cKO animals, more detail below. Finally, the authors do not convincingly demonstrate that the effects of knockouts are limited to MOC, not LOC neurons or with contributing effects from the middle ear muscles. Contributions from these other systems may indeed contribute to pathology and are also an important finding, but the language regarding the contributions of these other systems to pathology should be more clear.

Major:

• Evidence of MOC axon terminal degeneration: The transmission EM images showing degeneration of MOC efferent terminals onto OHCs in the null and ‘high’ lines requires at a minimum more detail of how many sections were analyzed because it is possible that the examples shown are simply from a sectioning plane below OHCs that does not happen to have an MOC terminal. While this is unlikely, more description of the process of examining multiple sections through OHCs is necessary to thoroughly eliminate the possibility that MOC neurons are healthy but in other sections, or MOC neurons are unhealthy and degenerating, or small, but present, which would add additional detail to the research. At present, only a single animal is used for each condition, with only a single example image shown and little detail regarding numbers, which is simply not enough evidence. More detail is required about the experiments that were performed. I would also recommend whole mount immunolabels with OC markers such as VAChT or CHAT or AChE to show a loss of MOC terminals across many more OHCs.

• Lack of effects of cKO on LOC efferents, middle ear muscles, central neurons. In these experiments LOC efferent neurons will also be disrupted at the same time as MOC neurons. The loss of LOC neurons may also contribute to the hearing deficits observed here. Is there a way to more clearly determine whether the LOC neurons are involved, or not? Similar to the suggestion above, cholinergic staining in the inner spiral bundle may assess LOC innervation. Also, more detailed analysis of ABR wave 1 amplitudes, latencies, etc may be used to assess changes to the function of the type I SGN independent of thresholds and OHC function. Some of the wording in the manuscript includes the possibility that LOC efferents play a role in auditory dysfunction (eg second to last sentence in the introduction) while other lines exclude LOC function (eg the final line of the introduction), and these should be clarified throughout.

Minor:

• Some references should be added:

o Intro line 13, LOC effects – add Groff and Liberman 2003 PMID: 14615429.

o Intro 15-16 MOC effects, add Romero and Trussell 2021 PMID: 34250904 for recent evidence of CN innervation of MOC neurons, also consider that MOC neurons receive inhibition: Torres Cadenas et al 2020 PMID: 31719165

o Intro lines 17-18 and discussion page 15 – consider recent work showing that the MEM reflex is more sensitive than previously appreciated, incorporate into work as necessary – 2 works by Valero PMID: 26657094 and PMID: 29598837 – does this influence your interpretation of your results?

o Intro first paragraph, last line – consider adding more recent works from the lab of Dr. Gomez Casati

o Intro page 4 – “intermediate part of the dorsal cochlear nucleus” is not a regularly used term, what does this refer to? Granule cell domains separating VCN and DCN? Granule cell domains within deep layers of the DCN? Other?

o Discussion first paragraph, last line on page 11 – add Frank et al 2023 for neurotransmitters in efferent neurons PMID: 36876911

o In multiple locations that discuss the effects of the MOC efferent system on development of central neurons Clause et al should be referenced PMID: 24853941

• Typos:

o Intro last line, ‘functionating’ to ‘functioning’

o Intro page 4, first full paragraph, line 6 – ‘evolution’ to ‘evolutionarily’

o Intro page 5, ‘previous to hearing functionality’ to ‘’prior to hearing functionality’

o Discussion first paragraph – an underscore is present instead of a space before ‘MOC’ in the middle of the paragraph

• Figure 1 legend – ‘n indicates the number of cochleae recorded’ should be included in the panel (D) text, not panel (E)

• The figure 1 colors look ok in a pdf, but they print poorly – consider using a different color for the Hoxb1null animals, or at least a different shade of blue. In addition, using different symbols in plots B and D for the different genotypes would aid readability.

• The text throughout figure 2 is too small

• Error bars appear to be either missing in plots 2 F and G, or are too small. I would expect them to be far larger than in plots C-E because there is obviously more spread in the data.

Reviewer #3: The study is important because HOXB1 mutations in humans cause hereditary hearing impairments.

Hoxb1 global mutations in mice also cause hearing loss, as earlier work by the authors has showed. The current study extends their previous work by using conditional mouse mutagenesis to pinpoint the cellular basis of Hoxb-1-linked hearing loss. The data strongly suggest that the interactions between olivocochlear motor neurons and outer hair cells during a critical postnatal period are crucial for the establishment of the cochlear amplification of sound. Major strengths of this work include: (a) four conditional mouse mutants are characterized, (b) Thorough electrophysiological analysis and high-resolution anatomical methods are employed, (c) Side-by-side comparison of conditional and null Hoxb1 mutant mice, and (d) the wide distribution of audiogram is nicely analyzed with hierarchical classification in all mouse mutant lines. A number of weaknesses we are also identified, as described below. Most (but not all) conclusions are supported by the data.

Major issues:

1. Title: Improve title of the paper because the causal role of Hoxb1 in generating the innervation phenotype is not clearly reflected.

The Introduction and Discussion can be condensed and improved.

2. Introduction: The authors are encouraged to condense and simplify the first paragraph of Introduction. Please, provide only the necessary information of cell types needed to understand the paper. Avoid the use of 10 acronyms (OHC, IHCs, VIII, CN, LL, LOC, MOC, IEE, MEM, PVCN). Are all these necessary to be mentioned for the reader to understand the paper? It seems that all the reader needs to know to understand the paper is MOC (motor neurons) and OHC (sensory neurons).

3. Discussion: Discussion is unnecessary long and needs a lot of work to be improved. This is important. The authors are strongly encouraged to reduce it to 3-4 pages maximum (current length is 6 pages), so the key take-home messages are succinct and easy for the reader to grasp. The first 2 pages of the Discussion add new information from the literature full of acronyms and anatomical terms. This part entitled “Development and function of efferent innervation…”is not necessary, and I am not sure what the discussed nAChR studies add to this paper. These 2 pages can be substituted with a short paragraph that tells the reader what is the knowledge gap, and what is the new information that this manuscript provides. The authors are encouraged to follow this guide: https://journals.plos.org/ploscompbiol/article?id=10.1371/journal.pcbi.1003453

4. A key message of the paper is that interactions between motor neurons and OHCs are crucial. Pre-synaptic and post-synaptic marker analysis would significantly strengthen this conclusion. Also, in the text and Abstract, the authors should make clear what do they think is the nature of this interaction. I presume it is a direct (chemical) synapse.

5. Essential to the conclusions of the paper is the detailed characterization of the conditional mouse mutants. The Nkx2.2 CKO line is partially characterized. The confirmation of reduced Hoxb1 mRNA expression in Nkx2.2Cre mice in ventral r4 is convincing, but the authors conclude, based on Tbx20, that a decreased population of LOC and MOC neurons is present. However, it remains unclear whether LOC and MOC neurons are normally born but fail to differentiate and extend axons and dendrites, or they are not born at all, or born and then die. Experiments with additional MN markers (GATA3, ChAT) or cell death markers could help distinguish between these possibilities. Moreover, in Figure 4, quantifications are needed of the number of Tbx20-expressing LOC and MOC neurons, similar to what the authors did in their previous 2013 work (PMID: 23408898). The issue of what exactly happens to LOC and MOC neurons is particularly important in light of the observation that audiograms in CKONkx2.2 line are split 50/50 in high- and low-threshold.

6. The manuscript mentions that OHCs degenerate (do not survive) in the the Nkx2.2 CKO line. Convincing data showing degeneration of these cells are not provided.

7. Data are not provided to show loss of Hoxb1 expression in sensory (inhibitory and excitatory) cochlear neurons in CKOAtoh1 and CKOPtf1a mice. This is very important in light of the absence of a phenotype in these animals (Fig. 2C-E).

8. Quantifications of data (as they authors did in PMID: 23408898) presented in Figure 3 and Figure 7 would strengthen the conclusions.

Minor issues:

In Figure 1, panel A, The authors should bring the schematic (WT only) from Figure 8. This would help the non-expert reader with auditory anatomy. It would also help communicate the key take-home message of this paper, i.e., the interaction between motor neurons (MOC) and OHCs.

In Introduction, last paragraph, please clarify that null phenotype is partially (not completely) reproduced in CKOr4 and CKONkx2.2 mutants. At the end of Results section, it is also important to clarify this as well.

**Have all data underlying the figures and results presented in the manuscript been provided?**

Reviewer #1: Yes

Reviewer #2: Yes

Reviewer #3: Yes

PLOS authors have the option to publish the peer review history of their article (what does this mean?). If published, this will include your full peer review and any attached files.

Reviewer #1: No

Reviewer #2: No

Reviewer #3: No

---

## [Decision Letter · Decision Letter 1]

22 Aug 2023

Dear Dr. Studer,

We are pleased to inform you that your revised manuscript entitled "Abnormal outer hair cell efferent innervation in Hoxb1-dependent sensorineural hearing loss" has been editorially accepted for publication in PLOS Genetics. Congratulations!

As you will see, all Reviewers state that your revised paper has been substantially improved and is now acceptable for publication. All three Reviewers initially requested additional evidence that the MOC efferent neurons are indeed absent (or reduced) in the various mouse lines you analyzed and suggested immunofluorescence experiments or tract tracing assays to provide a more robust confirmation of the results. However, in consideration of the fact that you no longer have the mice due to the refurbishment of your Animal Facility (which forced you to cryopreserve all of your mouse lines) and that you have now provided more detailed and extensive information regarding the TEM experiments, all Reviewers agree that the current evidence that the MOC terminals are reduced, or absent, is overall adequate.

Reviewer 1 suggests only a few very minor editorial changes, as detailed below, that we would like you to address promptly in order to send your manuscript to production.

Before your submission can be formally accepted and sent to production you will also need to complete our formatting changes, which you will receive in a follow up email. Please be aware that it may take several days for you to receive this email; during this time no action is required by you. Please note: the accept date on your published article will reflect the date of this provisional acceptance, but your manuscript will not be scheduled for publication until the required changes have been made.

Yours sincerely,

Licia Selleri

Academic Editor

PLOS Genetics

Hua Tang

Section Editor

PLOS Genetics

Comments from the reviewers (if applicable):

Reviewer's Responses to Questions

**Comments to the Authors:**

Reviewer #1: Using Hoxb1 deletion causes loss of MOC and LSO from IEEs. Absence of IEE lets to a shift in the ABR threshold. The paper has expanded and provides my previous suggestions. Very minor suggestions are attached.

Introduction

P3, line 7 from below: suggested changes: The assembly of central and peripheral auditory...

P4, line 11 from below: suggested changes: Besides sensory neurons, in ventral r4 (cite the work of Simmons et al., 2011

P6, line 3 from below: suggest adding the paper of Maricich et al, 2009

P11, line 14: suggest providing Simmone et al., 2011 that details the ipsilateral and contralateral input to MOC and LOC, respectively.

Reviewer #2: The authors have in part provided additional information to support their results, namely detailing that a number of mice and sections that were used to perform MOC counts in the various mutant mouse lines in TEM experiments. Their n is higher than I had initially thought. It is unfortunate that the mice are no longer available and so they can't perform additional experiments such as immunos, but I think that the weight of what they have performed and presented is enough to satisfy my doubts about whether MOC terminals to OHCs are indeed absent or reduced.

The authors have fully addressed my other concerns.

Reviewer #3: The authors have addressed my concerns. The revised manuscript is very much improved.

**Have all data underlying the figures and results presented in the manuscript been provided?**

Reviewer #1: Yes

Reviewer #2: Yes

Reviewer #3: Yes

PLOS authors have the option to publish the peer review history of their article (what does this mean?). If published, this will include your full peer review and any attached files.

Reviewer #1: No

Reviewer #2: No

Reviewer #3: **Yes: **Paschalis Kratsios

**Data Deposition**

http://datadryad.org/submit?journalID=pgenetics&manu=PGENETICS-D-23-00405R1

**Press Queries**

---

## [Editor Report · Acceptance letter]

31 Aug 2023

PGENETICS-D-23-00405R1 

Abnormal outer hair cell efferent innervation in *Hoxb1*-dependent sensorineural hearing loss 

Dear Dr Studer, 

We are pleased to inform you that your manuscript entitled "Abnormal outer hair cell efferent innervation in *Hoxb1*-dependent sensorineural hearing loss" has been formally accepted for publication in PLOS Genetics! Your manuscript is now with our production department and you will be notified of the publication date in due course.

With kind regards,

Lilla Horvath

PLOS Genetics

On behalf of:
